# Hemodynamic and Pulmonary Permeability Characterization of Hantavirus Cardiopulmonary Syndrome by Transpulmonary Thermodilution

**DOI:** 10.3390/v11100900

**Published:** 2019-09-27

**Authors:** René López, Rodrigo Pérez-Araos, Álvaro Salazar, Ana L. Ulloa, Cecilia Vial, Pablo A. Vial, Jerónimo Graf

**Affiliations:** 1Departamento de Paciente Crítico, Clínica Alemana de Santiago, Santiago 7650567, Chile; rene.lopezh@gmail.com (R.L.); rperez@alemana.cl (R.P.-A.); alvaro.salazarh@gmail.com (Á.S.); aulloa@alemana.cl (A.L.U.); 2Escuela de Medicina. Facultad de Medicina Clínica Alemana Universidad del Desarrollo, Santiago 7710162, Chile; pvial@udd.cl; 3Escuela de Kinesiología. Facultad de Medicina Clínica Alemana Universidad del Desarrollo, Santiago 7710162, Chile; 4Programa Hantavirus, Instituto de Ciencias e Innovación en Medicina (ICIM), Facultad de Medicina, Clínica Alemana Universidad del Desarrollo, Santiago 7590943, Chile; mcvialudd@gmail.com; 5Departamento de Pediatría, Clínica Alemana de Santiago, Santiago 7650567, Chile

**Keywords:** Hantavirus cardiopulmonary syndrome, Hantavirus pulmonary syndrome, transpulmonary thermodilution, pulmonary edema, Andes virus

## Abstract

Hantavirus cardiopulmonary syndrome (HCPS) is characterized by capillary leak, pulmonary edema (PE), and shock, which leads to death in up to 40% of patients. Treatment is supportive, including mechanical ventilation (MV) and extracorporeal membrane oxygenation (ECMO). Hemodynamic monitoring is critical to titrate therapy and to decide ECMO support. Transpulmonary thermodilution (TPTD) provides hemodynamic and PE data that have not been systematically used to understand HCPS pathophysiology. We identified 11 HCPS patients monitored with TPTD: eight on MV, three required ECMO. We analyzed 133 measurements to describe the hemodynamic pattern and its association with PE. The main findings were reduced stroke volume, global ejection fraction (GEF), and preload parameters associated with increased extravascular lung water and pulmonary vascular permeability compatible with hypovolemia, myocardial dysfunction, and increased permeability PE. Lung water correlated positively with heart rate (HR, *r* = 0.20) and negatively with mean arterial pressure (*r* = −0.27) and GEF (*r* = −0.36), suggesting that PE is linked to hemodynamic impairment. Pulmonary vascular permeability correlated positively with HR (*r* = 0.31) and negatively with cardiac index (*r* = −0.49), end-diastolic volume (*r* = −0.48), and GEF (*r* = −0.40), suggesting that capillary leak contributes to hypovolemia and systolic dysfunction. In conclusion, TPTD data suggest that in HCPS patients, increased permeability leads to PE, hypovolemia, and circulatory impairment.

## 1. Introduction

Orthohantaviruses, members of the *Hantaviridae* family, are rodent-born segmented negative strand RNA viruses [1,2]. Two main categories for orthohantavirus diseases have been described: hemorrhagic fever with renal syndrome in Asia and Europe caused by “old world” orthohantaviruses, and hantavirus cardiopulmonary syndrome (HCPS) in North and South America, caused by “new world” orthohantaviruses [3].

*Andes orthohantavirus* (ANDV) is the Orthohantavirus endemic to Chile and Argentina and its main reservoir is the long-tailed pygmy rice rat (*Oligoryzomys longicaudatus*) [4]. Humans are infected primarily by the inhalation of aerosolized excreta from infected rodents [5,6]. Additionally, ANDV is the only orthohantavirus known to be transmissible between humans [7,8,9,10]. Indeed, in cases of ANDV infection, sexual partners and contacts who slept in the same bed during the prodromal period have been associated with ten times more risk of becoming infected when compared to other household contacts [9,10].

The incubation period of ANDV varies from 7 to 49 days [11,12], followed by the cardiopulmonary phase, which evolves from a dry cough to respiratory failure due to capillary leak syndrome into the pulmonary interstitium. This is evidenced by chest radiographs showing peribronchial haze and Kerley’s B lines, that subsequently progress to alveolar flooding, developing HCPS, which also includes circulatory shock with myocardial depression [3,13].

Although the characteristic pathological feature of HCPS is the increased capillary permeability in lungs and depression of cardiac function, the detailed mechanism for such pathological changes remains unclear [14]. Hantaviruses primarily target endothelial cells, using β3 integrin receptor and protocadherin-1 for virus attachment and entry, triggering non-lytic endothelial cell dysregulation [15,16]. Data from some studies, mostly carried out in cell culture models, have proposed a multifactorial mechanism for hantavirus-induced capillary leakage, for example: damage of the endothelial cell barrier by up-regulated CD8+ T cell response [17]; over-expression of the vascular endothelial growth factor (VEGF) with degradation of VE-cadherin, an important adhesion molecule that modulates vascular permeability [18,19,20]; Ras homolog gene family member A (RhoA) protein activation for Andes virus N protein, and activated RhoA, who has been linked to endothelial permeability directed by thrombin, tumor necrosis factor alpha, histamine, bradykinin, and VEGF [21,22,23].

HCPS is one of the deadliest infectious diseases, with fatality rates of 35–40% [24,25,26,27]. Unfortunately, there are no drugs with proven efficacy for HCPS. Consequently, treatment is based on critical care support, including veno-arterial extracorporeal membrane oxygenation (VA-ECMO) [25]. The criteria to start VA-ECMO were reported by Crowley et al. in 1998—a cardiac index less than 2.0 L/min/m^2^ despite maximal support and at least one of the following conditions found to be associated with a 100% mortality rate: serum lactate greater than 4.0 mmol/L (normal range 0.0–2.2), a ratio of partial pressure of arterial oxygen to fraction of inspired oxygen (PaO_2_:FiO_2_) less than 60, or cardiopulmonary deterioration, such as arrhythmia or cardiac arrest [28]. These criteria highlight that the combination of hypodynamic shock (low cardiac index) and acute pulmonary edema (PE) characterize severe HCPS. It also stems from these criteria that hemodynamic monitoring is fundamental to assess the nature of circulatory impairment, to titrate resuscitative efforts and to decide VA-ECMO connection in a timely manner. Classically, hemodynamic monitoring has been performed by using the pulmonary artery catheter [25,28]. This device allows the measurement of cardiac output (CO), pulmonary artery pressure, central venous pressure (CVP), and pulmonary artery occlusion pressure as indicators of right and left ventricular filling pressures [25,28]. Since early 2000, hemodynamic monitoring can be performed by single indicator transpulmonary thermodilution (TPTD). This technique not only makes it possible to measure CO and volumetric cardiac preload, but also to assess PE at the bedside. In brief, TPTD provides two variables related to PE: extravascular lung water, which quantifies it, and the pulmonary vascular permeability index (PVPI), which differentiates increased permeability from hydrostatic PE [29,30]. Extravascular lung water index (EVLWi) obtained from TPTD has shown close correlation with the gold standard gravimetric method [30]. Moreover, in acute respiratory distress syndrome (ARDS) patients, an EVLWi higher than 15 mL/Kg has been independently associated with mortality [31,32].

Considering that pulmonary capillary leak and circulatory dysfunction characterize HCPS, TPTD appears to be a highly suitable method to assess them through the course of the disease. To date, a systematic description of TPTD measurements in HCPS patients has not been performed. The aim of this study is to describe the TPTD hemodynamic pattern in patients with HCPS admitted to a Chilean referral hospital due to *Andes orthonavirus* (ANDV). Additionally, we explored possible associations between PE and hemodynamic variables.

## 2. Methods

### 2.1. Study Design and Patients

This was an observational retrospective analytical study. The cohort was part of a prospectively obtained database by the Hantavirus Program from the Instituto de Ciencias e Innovación en Medicina de la Facultad de Medicina, Clínica Alemana—Universidad del Desarrollo, approved by our local IRB and ethics committee (ID19, September 29^th^ 2011). For this study, only patients admitted to the adult intensive care unit (ICU) of Clínica Alemana from 2011 to 2018 were considered.

The diagnosis of HCPS was suspected on clinical grounds and confirmed in all cases by quantitative enzyme-linked immunosorbent assay (ELISA), detecting ANDV-specific immunoglobulin M or detecting ANDV RNA by reverse-transcription polymerase chain reaction. Even though ELISA and PCR cross-reactivity between orthohantaviruses is well described, a substantial number of viruses have been sequenced in Chile and all have been ANDV [4].

Demographic, clinical, laboratory, and hospital mortality data were collected using a standardized case record form, and deidentified data was entered into a dedicated database.

In Clínica Alemana, all patients with a diagnosis of hantavirus diseases are admitted to the ICU, considering its unpredictable and potentially fatal course. Our protocol was to monitor all patients with hantavirus diseases presenting respiratory and/or circulatory failure by TPTD.

### 2.2. Methodological Aspects of Transpulmonary Thermodilution and Arterial Pulse Contour Analysis

The TPTD technique is based on the injection of a cold saline bolus close to the right atrium through a central venous catheter and recording the temperature change with time on the arterial side of the circulation, usually through a femoral artery catheter [33]. Aside from the known amount of thermal indicator given, the contour of the so generated temperature–time curve depends on blood flow and the volume of distribution of the indicator between the sites of injection and temperature measurement. An extensive mathematical analysis of this thermodilution curve makes it possible to compute blood flow (CO) and dissect the components of the volume of distribution of the indicator as follows (Figure 1):

(1) The maximal volume of blood in the four heart chambers or global end-diastolic volume (GEDV);

(2) The whole blood volume in the chest or intrathoracic blood volume (ITBV), which includes the GEDV and the pulmonary blood volume (PBV);

(3) The volume of fluid in the pulmonary interstitium and alveolar spaces where the thermal indicator can dissipate from the pulmonary capillaries or extravascular lung water (EVLW).

These variables can be combined to yield the following ones:

(1) The stroke volume (SV) is the ratio between CO and heart rate (HR);

(2) The global ejection fraction (GEF) is the ratio between SV and a fourth of the GEDV (considering the four cardiac chambers);

(3) The cardiac function index (CFI) is the ratio between CO and GEDV;

(4) The pulmonary vascular permeability index (PVPI) is the ratio between EVLW and PBV.

Cardiac output, SV, GEDV, and ITBV are anthropometrically normalized using the body surface area to yield the cardiac index (CI), stroke index (SI), GEDV index (GEDVi), and the ITBV index (ITBVi), respectively. Extravascular lung water is normalized to predicted body weight to yield the EVLW index (EVLWi) [33].

Other hemodynamic variables stem from systemic arterial pressure wave contour analysis: stroke volume variation (SVV) is the percentage change between maximal and minimal SV along 30 s, normalized by its mean value. In patients under controlled invasive mechanical ventilation (IMV) it represents cardiac preload dependency and may be used to predict the chance of increasing CO after a fluid challenge [34]. The maximal rate of arterial pressure increase per unit of time during systole (dPmax) has been proposed as an index of left ventricular function [35].

All the patients in this study were monitored using the PiCCO^TM^ system (Pulse Contour Cardiac Output, PULSION medical systems AG, Munich, Germany). This monitor computes all the parameters derived from the TPTD and arterial pressure waveform analysis described above. TPTD was performed in triplicate according to manufacturer’s recommendations, discarding measurements where CO differed more than 10% from the other two. Mean values of each variable were considered for analysis. On VA-ECMO the working principles of TPTD are violated, making it unreliable. Thus, only measurements performed before VA-ECMO connection were considered in these patients.

### 2.3. Variables of Interest

Variables obtained from TPTD can be categorized as follows:-“Classic hemodynamic”: HR, mean systemic arterial pressure (MAP), central venous pressure (CVP), CI, SI, and systemic vascular resistance index (SVRi), which is the ratio between the difference of MAP and CVP, and CI times 80;-Myocardial contractility: GEF, CFI, and dPmax;-Volumetric preload: ITBVi and GEDVi;-Fluid responsiveness predictor: SVV;-Pulmonary edema: EVLWi and PVPI.

All these variables were available for each patient studied.

All the data from all the thermodilutions performed in every patient were recorded. Values are expressed as mean ± standard deviation (SD).

### 2.4. Statistical Analysis

An initial descriptive analysis of the patients considered was performed. All the TPTD variables obtained from all patients were descriptively analyzed. To explore possible associations between PE and hemodynamic variables, linear regressions were performed between these variables and Pearson’s coefficients were obtained. To assess the behavior of hemodynamic variables according to PE severity, we dichotomized thermodilutions according to EVLWi; < 15 mL/Kg or EVLWi ≥ 15 mL/kg and compared them using the t-test or Mann–Whitney non-parametric test according to the distributions of the sample. Normality of data distribution was assessed using Kolmogorov–Smirnov’s test. Significance was defined as *p*-value < 0.05. Statistical analysis was performed using the IBM SPSS statistical package, version 20 (IBM Corp., Armonk, NY, USA).

## 3. Results

Eleven patients with confirmed HCPS monitored with TPTD were identified; all required vasopressors, five patients required IMV, three required VA-ECMO support after a failed trial of IMV and hemodynamic optimization, and three required non-invasive mechanical ventilation. All the patients survived and are still alive. Demographic, clinical, and laboratory characteristic of these patients are shown in Table 1.

A total of 133 TPTD were recorded. The median (IQR) number of measurements per patient was 11 [8,9,10,11,12,13,14,15]. Descriptive analysis with normal reference values for each variable is presented in Table 2. The main findings were low SI, GEF, and volumetric preload parameters associated with increased EVLWi and PVPI.

Correlations between PE and hemodynamic variables are summarized in Table 3. Among the significant correlations between EVLWi and hemodynamic variables, the most relevant positive correlations were with heart rate, CVP, and volumetric preload variables, and the most relevant negative correlations were with MAP and myocardial contractility (GEF and CFI) variables. Among the significant correlations between PVPI and hemodynamic variables the most relevant positive correlations were with HR and SVV; the most relevant negative correlations were with CI, volumetric preload, and myocardial contractility (GEF and CFI) variables.

A dichotomic analysis of TPTD measurements according to EVLWi showed that when PE is severe (EVLWi ≥ 15 mL/kg), HR and CVP are higher, while MAP, GEF, and CFI are lower (Table 4).

Additionally, we presented the time course of HR, SI, EVLWi, and PVPI in a representative patient that required VA-ECMO (Patient 2) and in a representative patient that did not require VA-ECMO support (Patient 4) (Figure 2). It is evident that there is an opposite trend of HR, SI, EVLWi, and PVPI as one patient aggravated, requiring VA-ECMO support in the end, and the other one recovered and was disconnected from IMV in the end.

## 4. Discussion

To the best of our knowledge, this is the first study that systematically describes the hemodynamic pattern, and quantifies pulmonary edema and lung permeability using transpulmonary thermodilution in a cohort of patients with HCPS requiring vasopressors. Only one previous case report suggested the usefulness of TPTD in HCPS [36]. The main findings of our study were a reduced SV compounded by low volumetric preload parameters (ITBVi and GEDVi) and a depressed myocardial contractility variable (GEF) associated with PE (increased EVLWi) with increased permeability (high PVPI). This profile is compatible with a combination of hypovolemia, systolic dysfunction, and increased permeability PE. Lung water content was correlated with tachycardia, arterial hypotension, and systolic dysfunction (GEF), suggesting that PE is linked to hemodynamic impairment. Pulmonary permeability was even better correlated with low volumetric preload (ITBVi and GEDVi), systolic dysfunction (GEF and CFI), hypodynamic state (CI), and tachycardia, suggesting that pulmonary capillary leak contributes to hypovolemia and systolic dysfunction. Along these lines, individual measurements with high EVLWi (≥15 mL/Kg) were also associated with arterial hypotension and reduced systolic performance (GEF and CFI).

Our findings perfectly agree with those of the only previous systematic description of HCPS hemodynamics and lung permeability assessment, by Hallin et al. in 1996, although through quite different techniques. Using the pulmonary artery catheter, chest X-rays, and the ratio of tracheal fluid to plasma protein and albumin concentration, they documented PE with low pulmonary artery occlusion pressure and high PE fluid to the plasma protein concentration ratio in the context of circulatory shock with hypovolemia at presentation in patients with HCPS due to the sin nombre virus [37]. Once fluid resuscitated, SI remained low, suggesting that myocardial depression was also present in these patients [37]. We were able to quantify this PE and document the same high pulmonary permeability with TPTD in our patients. We also observed hypovolemia through volumetric preload assessment and a dynamic predictor of fluid responsiveness. Interestingly, we both observed that stroke index was a better indicator of circulatory stress than CI, due to compensatory tachycardia. We also observed a trait of myocardial depression, mainly through the GEF, an index that relates stroke volume to volumetric preload, but we were not able to document low stroke volume with high volumetric preload due to our restrictive approach to fluid resuscitation.

Controlled mechanical ventilation, particularly with positive end-expiratory pressure (PEEP), can produce complex genuine or artifactual hemodynamic effects. The most important genuine effects are a reduction in venous return and preload, coupled with an increase in afterload to the right heart that may result in reduced SV and CO [38]. Importantly, IMV does not affect myocardial contractility. Pressure indicators of cardiac preload (CVP and pulmonary artery occlusion pressure) may artifactually increase with PEEP application [39,40]. Among TPTD variables, volumetric preload parameters, CO and SV, are devoid of artifacts induced by increased intrathoracic pressure and may truly decrease with PEEP [39]. In contrast, PEEP may falsely reduce the thermodilution EVLW measurement by excluding portions of the lung from thermal indicator distribution through pulmonary capillary compression [40]. The magnitude of these effects of IMV on hemodynamics depend mainly on intravascular volume status, level of PEEP applied, and its effects on lung parenchyma [38]. Dynamic predictors of fluid responsiveness, such as SVV, depend mainly on intravascular volume status and tidal volume set on the ventilator [34]. In our patients it was difficult to separate the hemodynamic effects of IMV from those of the disease process itself. A rough comparison of the first TPTD variables between the nine patients that received IMV (with a mean PEEP of 10 cmH_2_O) and the three who did not insinuates a higher CVP and lower volumetric preload and SI in the first group, suggesting some hemodynamic impact of IMV. Nevertheless, EVLWi was similarly high and PVPI was higher in the IMV group, suggesting a greater impact of the HCPS on permeability in these patients. Furthermore, the different trend of HR, SI, EVLWi, and PVPI between the two patients on IMV depicted on Figure 2 suggests a role of the disease process itself on the course of hemodynamic and PE variables.

A novel observation of our report is the association of PE and increased permeability to circulatory failure with elements of both hypovolemia and myocardial depression. The association of capillary leak and hypovolemia is self-explanatory, though in contrast to “old world” orthohantaviruses, in “new world” orthohantaviruses the leak is most prominent in the lung vasculature. The association of capillary leak and myocardial depression is more intriguing. Myocardial edema is a relatively newly recognized entity and it has been shown that heart function can be significantly compromised with only small increases in myocardial interstitial fluid [41]. One histopathologic study showed that patients who died from HCPS had more interstitial myocardial edema than other critically ill patients who died [42]. Putting together these scattered data, one could speculate that capillary leak could also affect the myocardium in HCPS patients, leading to transient systolic dysfunction.

On an individual patient basis, it is interesting to see that stroke index behaves as the mirror image of pulmonary permeability, lung edema, and compensatory tachycardia along the course of the disease (Figure 2). The course of such derangements seems to also anticipate the severity of the disease and could eventually provide additional criteria for timely VA-ECMO support.

The weaknesses of our study are its retrospective nature, the small number of patients, and the high variability in the number of measurements between patients. Its strengths are the large number of measurements collected for analysis and the inclusion of patients with different severities in terms of supportive therapy requirements.

In conclusion, transpulmonary thermodilution provides a bedside characterization of the hemodynamic profile in HCPS consistent with the pathophysiology of the disease and older descriptions using different diagnostic tools. Increased permeability pulmonary edema may be pathophysiologically linked to hypovolemia and eventually to myocardial depression. This TPTD pattern with evidence of reduced preload, myocardial contractility, and increased permeability PE, though highly suggestive of HCPS in the proper epidemiological context, cannot be considered specific, as sepsis can produce all three of them. In fact, ARDS is characterized by increased permeability PE [29,32] and sepsis is one of its main causes [43]. More studies are needed to determine if specific transpulmonary thermodilution patterns can be useful to anticipate the need for VA-ECMO support.

## Figures and Tables

**Figure 1 viruses-11-00900-f001:**
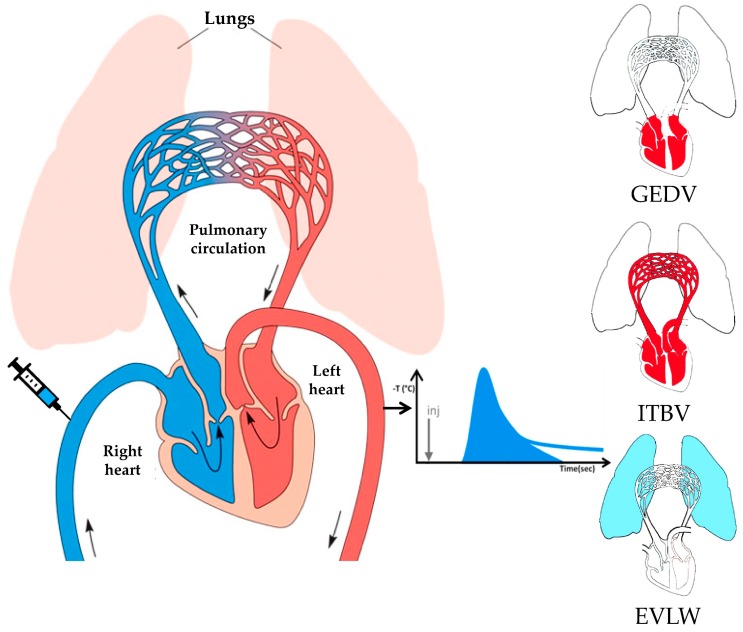
Schematic representation of intrathoracic volumes assessment by transpulmonary thermodilution (TPTD). The technique is based on central venous injection of cold saline and its detection close to the aorta usually via a femoral artery catheter. The temperature–time (thermodilution) curve generated after passage of the indicator through the central circulation and the lungs is illustrated on the right side of the site of detection. The three main volumes computed are depicted on the right side of the thermodilution curve. Global end-diastolic volume (GEDV) consists of left and right, atrial and ventricular end-diastolic volumes (top). Intrathoracic blood volume (ITBV) is the GEDV plus the pulmonary blood volume (middle). GEDV and ITBV are volumetric preload parameters. Extravascular lung water (EVLW), a pulmonary edema quantification index, is the volume of fluid in the pulmonary interstitium and alveolar spaces accessible to thermal diffusion from the pulmonary capillaries (bottom). Figure modified from Non-Communicable Diseases, Emergency Care and Mental Health module of the HEAT Programme, Module: 1. Cardiovascular Diseases, 1.2 Anatomy and physiology of the heart. An OpenLearn Create chunk used/reworked by permission of The Open University © (2011), (https://www.open.edu/openlearncreate/mod/oucontent/view.php?id=287&printable=1&extra=thumbnailfigure_idm4070912).

**Figure 2 viruses-11-00900-f002:**
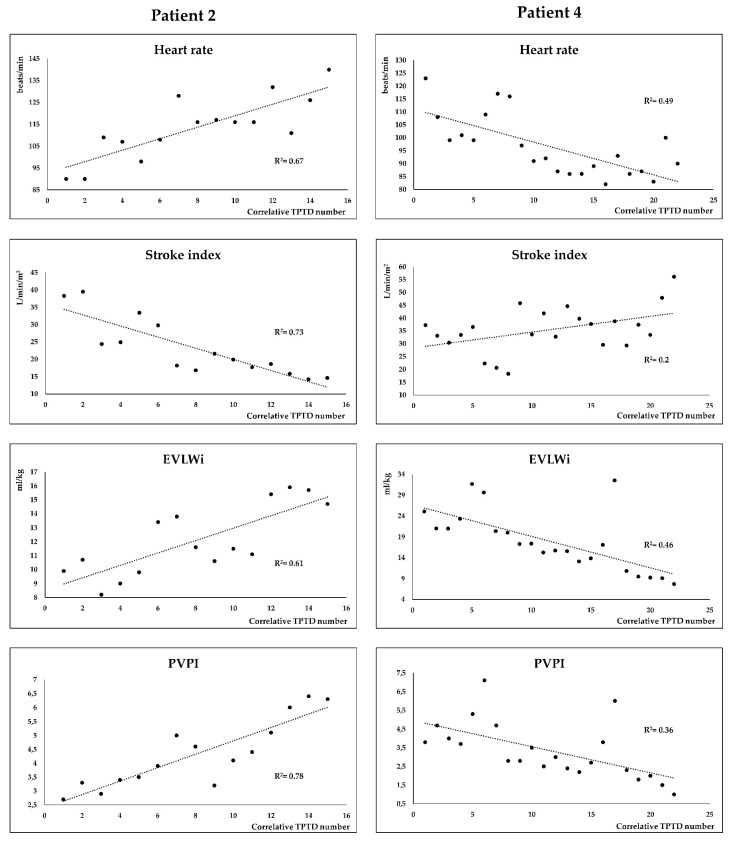
Time course of heart rate (HR), systolic index (SI), extravascular lung water index (EVLWi) and pulmonary vascular permeability index (PVPI) in a representative patient that required veno-arterial extracorporeal membrane oxygenation (VA-ECMO) (Patient 2) and in a representative patient that did not require VA-ECMO support (Patient 4).

**Table 1 viruses-11-00900-t001:** Demographic and clinical, characteristics of patients with hantavirus cardiopulmonary syndrome (HCPS) monitored with transpulmonary thermodilution (TPTD). Sequential organ failure assessment (SOFA), acute physiology and chronic health evaluation (APACHE II), veno-arterial extracorporeal membrane oxygenation (VA-ECMO), intensive care unit (ICU), length of stay (LOS).

Characteristic	Value
Male, N (%)	7 (64)
Age, years, median (range)	29 (15–59)
SOFA score, points, median (range)	9 (4–14)
APACHE II score, points, median (range)	10 (5–30)
Invasive mechanical ventilation, N (%)	8 (73)
Inotropic drugs, N (%)	8 (73)
VA-ECMO, N (%)	3 (27)
ICU-LOS, days, median (range)	6 (4–18)
Hospital-LOS, days, median (range)	12 (4–87)

**Table 2 viruses-11-00900-t002:** Descriptive analysis of transpulmonary thermodilution (TPTD) variables. Data are expressed as median and interquartile range (IQR). Reference values are given for each variable. dPmax is the maximal change of arterial pressure per second.

Transpulmonary Thermodilution Variables	Median (IQR)	Reference Range
**Classic hemodynamic**		
Heart rate, beats/min	99 (90–109)	60–100
Mean arterial pressure, mmHg	82 (75–89)	70–90
Central venous pressure, mmHg	8 (4–11)	6–12
Cardiac index, L/min/m^2^	3.1 (2.5–3.8)	3.0–5.0
Stroke index, mL/m^2^	34 (26–41)	40–60
Systemic vascular resistance index, dyn·sec·cm^−5^·m^−2^	1880 (1546–2364)	1700–2400
**Myocardial contractility**		
Global ejection fraction, %	24 (21–27)	25–35
Cardiac function index, 1/min	5.6 (5.0–6.3)	4.5–6.5
dPmax, mmHg/s	950 (762–1094)	900–1200
**Cardiac preload**		
Intrathoracic blood volume index, mL/m^2^	667 (553–790)	850–1000
Global end diastolic volume index, mL/m^2^	538 (442–635)	680–800
**Fluid responsiveness predictor**		
Stroke volume variation, %	13 (8–17)	≤10
**Pulmonary edema**		
Extravascular lung water index, mL/Kg	13.1 (10.2–17.3)	3.0–7.0
Pulmonary vascular permeability index, dimensionless	3.2 (2.7–4.7)	1.0–3.0

**Table 3 viruses-11-00900-t003:** Pearson correlations between pulmonary edema (PE) and hemodynamic variables. PE variables are the extravascular lung water index (EVLWi) and pulmonary vascular permeability index (PVPI). Hemodynamic variables are categorized as classic hemodynamic, myocardial contractility, cardiac preload, and fluid responsive prediction variables. dPmax is the maximal change of arterial pressure per second.

Transpulmonary Thermodilution Variables	EVLWi	*p* Value	PVPI	*p* Value
**Classic hemodynamic**				
Heart rate, beats/min	0.20	0.02	0.31	<0.01
Mean arterial pressure, mmHg	−0.27	<0.01	−0.10	0.26
Central venous pressure, mmHg	0.23	0.01	0.26	<0.01
Cardiac index, L/min/m^2^	0.02	0.79	−0.49	<0.01
Stroke index, mL/m^2^	0.06	0.51	−0.12	0.17
Systemic vascular resistance index, dyn·sec·cm^−5^·m^2^	−0.18	0.04	0.38	<0.01
**Myocardial contractility**				
Global ejection fraction, %	−0.36	<0.01	−0.40	<0.01
Cardiac function index, 1/min	−0.25	<0.01	−0.20	0.02
dPmax, mmHg/s	0.12	0.18	0.16	0.08
**Cardiac preload**				
Intrathoracic blood volume index, mL/m^2^	0.21	0.01	−0.48	<0.01
Global end diastolic volume index, mL/m^2^	0.21	0.01	−0.48	<0.01
**Fluid responsiveness predictor**				
Stroke volume variation, %	0.10	0.28	0.22	0.01

**Table 4 viruses-11-00900-t004:** Dichotomic analysis of transpulmonary thermodilution (TPTD) measurements according to extravascular lung water index (EVLWi) with a threshold of 15 mL/Kg. dPmax is the maximal change of arterial pressure per second. Values are mean with standard deviation (SD) in parenthesis. Comparisons were made using the t-test.

Hemodynamic Variables	EVLWi < 15 mL/kg	EVLWi ≥ 15 mL/kg	*p*-Value
**Classic hemodynamic**			
Heart rate, beats/min	96 (18)	103 (14)	0.02
Mean arterial pressure, mmHg	84 (10)	79 (9)	<0.01
Central venous pressure, mmHg	6 (4)	10 (5)	<0.01
Cardiac index, L/min/m^2^	3.2 (0.9)	3.1 (0.9)	0.74
Stroke index, mL/m^2^	34 (9)	35 (30)	0.75
Systemic vascular resistance index, dyn·sec·cm^−5^·m^2^	2093 (702)	1915 (620)	0.15
**Myocardial contractility**			
Global ejection fraction, %	25 (4)	22 (5)	<0.01
Cardiac function index, 1/min	5.9 (1.1)	5.4 (0.9)	0.01
dPmax, mmHg/s	925 (223)	1007 (304)	0.12
**Cardiac preload**			
Intrathoracic blood volume index, mL/m^2^	680 (172)	735 (222)	0.12
Global end diastolic volume index, mL/m^2^	546 (138)	588 (177)	0.14
**Fluid responsiveness predictor**			
Stroke volume variation, %	13 (6)	14 (7)	0.68
**Pulmonary edema**			
Extravascular lung water index, mL/Kg	11.1 (2.2)	20.0 (4.3)	<0.01
Pulmonary vascular permeability index, dimensionless	3.1 (0.9)	5.1 (1.7)	<0.01

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
