# Peer review of "Hemodynamic and Pulmonary Permeability Characterization of Hantavirus Cardiopulmonary Syndrome by Transpulmonary Thermodilution"

_viruses, 2019, doi:10.3390/v11100900_

Round 1

Reviewer 1 Report

The authors analyzed the cause of pulmonary edema in HPS patients due to Andes virus infection by using TPTD. TPTD is a relatively new tool for haemodynamic monitoring. It must be a useful tool for treatment of HPS, , which shows acute pulmonary edema with high mortality. However, no systematic data of HPS patients by TPTD has not been reported. Finally, authors concluded association of pulmonary edema and increased permeability. 

This paper gives much information on the treatment of HPS. Data obtained from TPTD can be used as an indicator or treatment , such as ECMO use. Although data obtained from TPTD varies among patients, continuous monitoring by minimally invasive method is also helpful to clear individual difference. 

Myocardial depression might be caused by orthohantaviruses. However mechanisms is not clear.  It is interesting that this study showed the possibility of myocardial edema due to capillary leakage.

Author Response

We appreciate the encouraging comments on our work and wish the changes made contribute to make  it a better communication.

Changes in the text dealing with reviewer observations were highlighted using the "Track Changes" function in Microsoft Word in the new version of the manuscript.

Reviewer 2 Report

Lopez et al investigated the hemodynamic and pulmonary permeability characterization of 11 patients with HCPS by transpulmonary thermodilution (TPTD). Authors showed that reduced stroke volume, global ejection fraction and preload parameters were associated with increase of extravascular lung water and pulmonary vascular permeability. TPTD data suggested that increased permeability leads to pulmonary edema, hypovolemia and circulatory impairment in HCPS patients. This study was interested, however some concerns included.

Do TPTD data suggest similar results in other diseases which had pulmonary edema or left ventricular dysfunction? How was the GEF, CFi, dPmax, ITBVi, GEDVi, SVV, EVLWi and PVPi obtained form TPTD data? Did authors calculate from TPTD data or volume curve? Please show the representative image of these data or describe the method in detail. Did use of VA-ECMO or invasive mechanical ventilation affect the results? Did authors change the treatment by TPTD data? Are there any specific parameters with HCPS?

Author Response

Changes in the text dealing with reviewer observations were highlighted using the "Track Changes" function in Microsoft Word in the new version of the manuscript.

The reviewer raises a number of interesting questions which were address by grouping them:

1) How was the GEF, CFi, dPmax, ITBVi, GEDVi, SVV, EVLWi and PVPi obtained from TPTD data? Did authors calculate from TPTD data or volume curve? Please show the representative image of these data or describe the method in detail.

We added a section within Methods dealing with "Methodological aspects of transpulmonary thermodilution" where we broadly explain the methodology of transpulmonary thermodilution together with the physiological meaning of each one of the primary and derived parameters. In addition we included a figure to further clarify these concepts.

2) Did use of VA-ECMO or invasive mechanical ventilation affect the results?

At the end of the section on "Methodological aspects of transpulmonary thermodilution" we added a paragraph where we explain that TPTD measurements were no longer performed once patients were connected to VA-ECMO due to the unreliability of thermodilution under VA-ECMO.

Mechanical ventilation may have extensive effects on hemodynamics that are difficult to dissect from the effects of HCPS itself. A paragraph explaining the circulatory effects of IMV, as well as potential artifactual effects on monitored variables was added to the discussion. A rough comparison of the initial TPTD measurements in patients that received IVM and those that did not suggests mixed effects of IMV and the disease process. A paragraph on these preliminary observations was also added to the discussion.

3) Do TPTD data suggest similar results in other diseases which had pulmonary edema or left ventricular dysfunction? Are there any specific parameters with HCPS? Did authors change the treatment by TPTD data?

These questions are more difficult to answer.

A paragraph indicating that a combination of hypovolemia, systolic dysfunction and increased permeability pulmonary edema may be suggestive of HCPS in an adequate epidemiological context was added to the discusion; but we also warn that sepsis could well produce the same combination of findings  through septic shock, septic myocardial depression and acute respiratory distress syndrome.

We used TPTD variables to assess the response to different interventions such as cautious volume loading, vasopressors, inotropic drugs, sedatives, adjustments to the mechanical ventilator settings and continuous veno-venous hemofiltration as well as to prepare for ECMO connection when all these interventions failed to restore homeostasis. Due to the retrospective nature of our study and the small number of patients included it is difficult to present this information in a meaningful manner. We also think that such an effort could deviate the attention from our main objective which was to present a quantitative description of the effects of HCPS on circulation and pulmonary permeability through TPTD.

Round 2

Reviewer 2 Report

I am satisfied with authors' responses.